# Thorny Roses: Investigating the Dual Use Dilemma in Natural Language Processing

**Lucie-Aimée Kaffee[1], Arnav Arora[2], Zeerak Talat[3], Isabelle Augenstein[2]**
[1]Hasso Plattner Institute, Germany, [2]University of Copenhagen, Denmark
[3]Mohamed Bin Zayed University of Artificial Intelligence, United Arab Emirates
lucie-aimee.kaffee@hpi.de, aar@di.ku.dk, z@zeerak.org, augenstein@di.ku.dk

## Abstract

Dual use, the intentional, harmful reuse of technology and scientific artefacts, is an ill-defined problem within the context of Natural Language Processing (NLP). As large language models (LLMs) have advanced in their capabilities and become more accessible, the risk of their intentional misuse becomes more prevalent. To prevent such intentional malicious use, it is necessary for NLP researchers and practitioners to understand and mitigate the risks of their research. Hence, we present an NLP-specific definition of dual use informed by researchers and practitioners in the field. Further, we propose a checklist focusing on dual-use in NLP, that can be integrated into existing conference ethics-frameworks. The definition and checklist are created based on a survey of NLP researchers and practitioners.[1]

## 1 Introduction

As usage of NLP artefacts (e.g., code, data, and models) increases in research and commercial efforts, it becomes more important to examine their rightful use and potential misuse. The Association for Computing Machinery code of ethics[2], which has been adopted by ACL[3], states that *"[c]omputing professionals should consider whether the results of their efforts (...) will be used in socially responsible ways"*. New regulation in the European Union also requires producers of LLMs to outline foreseeable misuses and mitigation techniques (EU, 2023). However, without guidance on how to assess the impacts on society, researchers and practitioners (henceforth professionals) are ill-equipped to consider social responsibility in deciding on tasks to work on or managing resulting

research artefacts. This is reflected in contemporary ethical review processes which emphasise the impacts of research on individual subjects, rather than the wider social impacts of conducted research.

While very few research projects have malicious motivations, some are reused to harm any but particularly marginalised groups of society. This presents a crucial gap in the ethics of artificial intelligence and NLP on malicious reuse, or *dual use*, which has been particularly absent in literature.

Concerns of dual use of Artificial Intelligence (AI) have been discussed by prior work (e.g., Shankar and Zare, 2022; Kania, 2018; Schmid et al., 2022; Urbina et al., 2022; Ratner, 2021; Gamage et al., 2021). However, NLP technologies are rarely included in such considerations. As LLMs are being incorporated into a wide range of consumer-facing products, the dual use considerations become increasingly critical for research and deployment. For instance, a online mental health services company has used ChatGPT to respond to people seeking advice and aid, raising concerns of informed consent, and impact on vulnerable users (Biron, 2023). To address such misuse of LLMs, it is crucial to support researchers in making decisions to limit the dual use of their work.

We propose a checklist that can be implemented by researchers to guide them in the decision making. If integrated into a paper, the checklist further serves to elucidate the considerations made and approaches taken to mitigate dual use to other professionals. We keep the checklist concise, such that it can readily be integrated into existing checklists for ethics used by conferences within NLP. The checklist is developed based on a mixed-methods survey. In the survey of NLP professionals, we find that the majority of our respondents do not spend a significant amount of time considering dual use

---

[1] ○ We make the survey and checklist available at: https://github.com/copenlu/dual-use

[2] Association for Computing Machinery Code of Ethics

[3] Association for Computational Linguistics Code of Ethics

concerns arising from their work and that institutional support often is inadequate.

In summary, our paper contributes: (a) a definition of dual use for NLP artefacts; (b) a survey of NLP professionals documenting their concerns of misuse of NLP artefacts; and (c) a checklist and outlines of methods for mitigating risks of dual use.

## 2 Dual Use in NLP

NLP artefacts, such as LLMs, can have severe impacts on society, even when they are not developed with malicious intent. Addressing the issue of malicious use, the European Union defines dual use as: *"goods, software and technology that can be used for both civilian and military applications".*[4] However, similarly to many other definitions of dual use, this definition only transfers to a subset of NLP projects. From the field of neuroscience, Mahfoud et al. (2018) suggest that the current military use and export control lens for dual use is not adequate for scientists and policy-makers to anticipate harms and recommend investigating the political, security, intelligence, and military domains of application. Drawing from dual use research in life sciences, Koplin (2023) highlight the need for considerations in AI in light of NLP models as sophisticated text generators.

We argue that the field of NLP is at a crossroad – on one hand, artefacts have become more accessible. On the other, the increased access to resources limits the control over their use. While availability and open access can facilitate reproducible science, there is a need to develop practices that prevent the malicious use of artefacts. Discussing the social impacts of NLP, Hovy and Spruit (2016) describe dual use as *"unintended consequences of research".* However, this definition leaves malicious reuse out of scope. Resnik (2009) argue that dual use definitions should neither be too narrow or too broad in scope, as that results in limiting what can be considered or make mitigating risks of dual use unmanageable. Forge (2010) further posit that the context of dual use is critical in determining its implications and context-specific threats and risks should be specified.

Therefore, in our work, we present a NLP specific definition of dual use, and an associated checklist that is based on the insights from our survey of

the field. Our definition is founded in a classification of harms (Weidinger et al., 2021).

### 2.1 Defining Dual Use in NLP

We conduct a survey of NLP professionals (see Section 3.2) and found that participants were interested in a precise definition that communicates clearly to professionals who are unfamiliar with dual use. We therefore formulate a NLP specific definition of dual use that is informed by our survey results and prior work. Our definition centres the intent of work being created, and the intent with which it is reused. While our definition is informed by our insights into the NLP research community, it may be applicable to other fields. We advise that researchers from other fields carefully evaluate whether their field would require any changes to the definition to capture field-specific challenges.

**Definition** *We understand **dual use** as the **malicious reuse** of technical and research artefacts that were developed without harmful intent. **Malicious reuse** signifies applications that are used to harm any, and particularly marginalised groups in society, where **harm** describes the perceived negative impacts or consequences for members by those groups.* That is, dual use describes the intentional and malicious re-use of for a harmful (secondary) purpose besides its primary application area. To define harm in the context of dual use, we differentiate between sanction and violence, where our definition of harm focuses on violence rather than sanction. Briefly, *sanction* can be described simply as violence that is perceived as justified by entities which hold power (e.g., penalties by a government for breaking regulations on technological use), whereas *violence* can simply be described as harms without leave (e.g., breaking regulation which causes harm) (Foucault, 2012; Bentham, 1996). The primary mode through which these differ is by how power is distributed.

**Exclusions** We exclude from our definition the *unintended secondary* harms from research artefacts. For instance, our definition does not address the gendered disparities produced in machine translation, as the reproduction of patriarchal values is a secondary harm that arises from the social context and is reflected as model and dataset biases (Vanmassenhove et al., 2018; Hovy et al., 2020). Similarly, we exclude from our definition tasks where the primary uses are malicious or harmful, e.g. developing research artefacts to predict recidivism

---

[4]`https://policy.trade.ec.europa.eu/help-exporters-and-importers/exporting-dual-use-items_en`

rates, as the *primary* use-case is harmful (Leins et al., 2020).

## 3 Survey of NLP Professionals

In order to develop a checklist, we survey members of the NLP community on their stances towards misuse of research artefacts through an anonymous online form. Michael et al. (2022) conduct a survey of NLP professionals' perspectives on general topics, e.g. potential decline in industry interest and promising research directions, dual use is notably absent from consideration. Indeed, only one question alludes to the issue: *"Is it unethical to build easily-misusable systems?"* to which 59% respond in the affirmative, indicating that professionals are responsible for minimising misuse within their practice. We similarly conduct an anonymous survey of NLP professionals, seeking detailed insights into their views on dual use concerns. Our survey sought to elicit participants understanding of dual use, their perception of harms arising from it, and how to mitigate such harms.

Our adapted survey questions were shaped by feedback on a preliminary version shared researchers within the same group, at different levels of seniority and discussions with colleagues in academia and industry. The survey was open to the NLP community, widely, from April until the end of June 2022 and advertised on Twitter, professional mailing lists and relevant Reddit communities.[5] The survey was conducted on the LimeSurvey[6] platform, and approved by the University of Copenhagen ethics board under number 504-0313/22-5000. The survey contained a total of 23 questions, which consist of a mixture of multiple-choice questions, free-form text answers, and Likert scales (see Appendix B for the survey setup). We inductively code all free-form text answers (see Table 2 for the codebook). For our analysis, we discard all partial responses, resulting in $n = 48$ complete responses out of 256 participants. We discarded partial responses as they were mostly limited to the participant demographics questions (the first part of the survey).

### 3.1 Demographics

The majority of participants work in academia (62.5%) across career stages (see Figure 1), with

---

[5]Reddit communities: *r/nlproc*, *r/MachineLearning*, *r/LanguageTechnology*

[6]https://www.limesurvey.org/

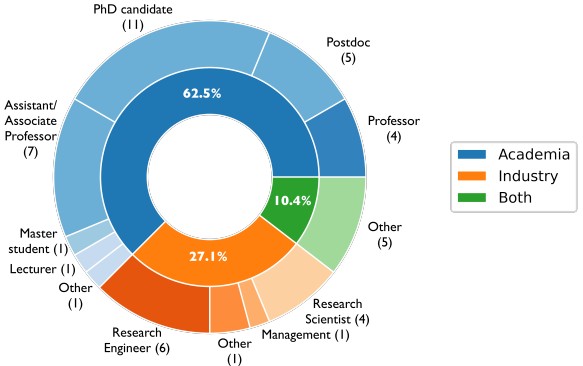

Figure 1: Survey respondents' occupation.

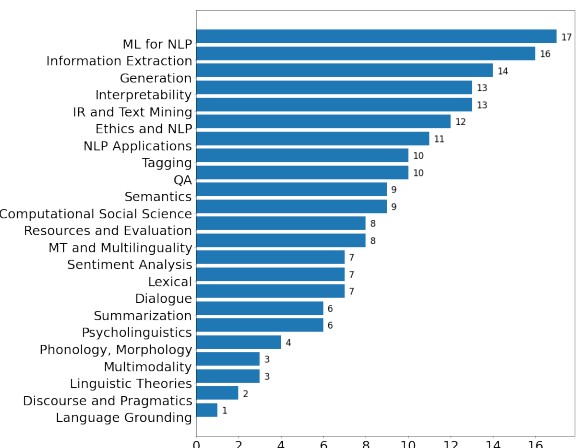

Figure 2: Respondents' area of work.

PhD students comprising the largest group of participants. Of participants from industry, most work as Research Scientists. Participants are primarily based in Europe and North America with only three participants based in Asia, and one in Africa. This limits the validity of our results to Europe and North America (see Section 6 for further discussion).

**Areas of work** Participants are asked to select their areas of work and the task they work on, using the ACL 2020 classification of research areas.[7] Participants can select multiple areas and were asked to add at least one task per area. Their responses of work cover a wide range of topics demonstrating a high degree of diversity of areas of work and expertise (see Figure 2 for the distribution of responses). This is further reflected by participants working on 180 unique tasks. Using participants' input, we identify potential harms for each task in Section 3.3.

---

[7]https://acl2020.org/calls/papers/#submissions

## 3.2 Definition of Dual Use

As developing a NLP specific definition of dual use was one of the primary goals of conducting our survey, we presented participants with the following definition of dual use and a free-form text field to respond:

> "Dual use describes any task that can be intentionally used for a harmful (secondary) purpose besides its main application area."

While a slight majority of participants (56%) agree with the definition, their responses revealed that a majority found the definition to be too vague. For instance, participants requested clearer definitions of *"task"* and *"harmful"* such that the definition would be clear to people unfamiliar with dual use. Further, participants emphasised the need to consider the power relations afforded by the definition. Given the participant responses, we adapt our definition of dual use to the one presented in Section 2.1.

## 3.3 Perception of harm

Given that NLP professionals shape and implement NLP technology, the survey explores participants' perception of harms in the field of NLP (see Figure 3). When we asked about the risks of dual use, a majority of participants found NLP overall vulnerable, see Figure 3a. Indeed, we find that a majority of participants (77%) perceive dual use as somewhat or very important to their research, however only a minority (46%) of participants reported to think about it often or always. This demonstrates that while professionals hold a general concern, it is not frequently considered in NLP projects.

The survey asks participants how they approach dual use considerations in their research. The answers cover a broad range of concerns. For instance, a large share of participants considered how scientific artefacts may be reused: *[P. 11]: "How could this tool be used for harmful purposes?"*. Participants also raised questions that indicate impacts on what they work on, e.g., *"Should I build this?"*. Participants were further concerned about the application area of their work:

> [P. 28]: *"How could a corporation or government use this to surveil people?"*

The wide range of questions and concerns raised by participants provided insight into the nuanced consideration of dual use by NLP professionals and serve as a basis for the checklist (see Section 5).

**Dual Use in the Research Workflow** In efforts to understand the stages in which dual is considered, the participants select stages of their research project in which they consider the risks of misuse (for an overview of responses, see Figure 4). Although 58.3% of respondents selected that they did not find the proposed stages fitting to when they considered misuse, they also selected stages that most closely fit the stage in which they consider misuse. This indicates that further specificity is required in identifying which stages such considerations occur. For the respondents who select one or more stages for considering dual use, the discussions predominantly happen in collaborative parts of the research process, i.e. ideation and writing phases. Less consideration is given after papers have been submitted and in the post publication processes. While it is important to consider dual use in the early phases, i.e. when selecting tasks to work on, potential harms can often be more clearly identified once scientific artefacts have been produced. Our checklist (see Section 5) therefore seeks to aid professionals in considering dual use in the later stages of research.

**Harms by NLP tasks** We define the areas of work in the survey using the areas of work described in the ACL 2020 call for papers.[8] We ask participants to identify the areas and tasks that they work on to obtain expert views on dual use in these fields. For each task they are asked to provide (1) harmful use cases and (2) how vulnerable they are to misuse. We then codify the harms described, assigning one code per described harm (see Figure 5). Intentional manipulation, e.g. dis/misinformation and attempted polarisation of users is frequently listed as a potential harm. The use language models for such tasks has previously been discussed as *Automated Influence Operations* in detail by Goldstein et al. (2023). Further frequently identified harms include oppressing (marginalised) groups of society, e.g. using research artefacts to produce discriminatory outcomes, and surveillance by government or corporate entities. Other concerns centre around reusing NLP technologies for criminal purposes (e.g., fraud), ethics washing, plagiarism, censorship, and military applications.

Participants were also asked to score how vulner-

---

[8]ACL 2020 Call for Papers.

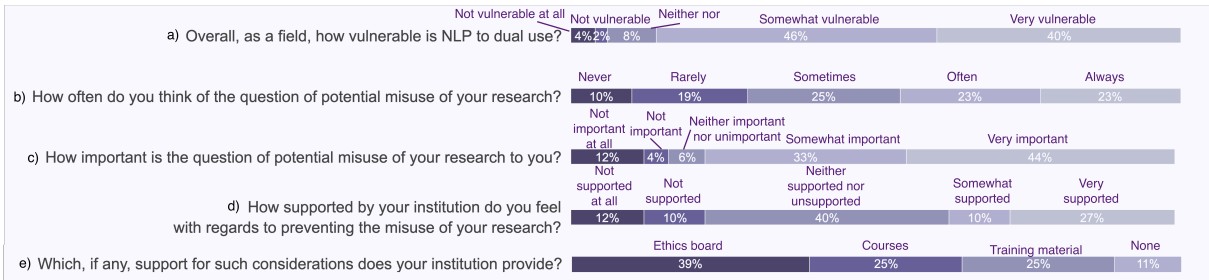

Figure 3: Likert scales of responses on *perceptions of harm* and institutional support for *preventing misuse*.

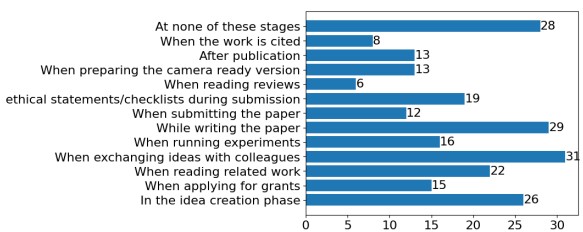

Figure 4: Participant responses to which stage of a project they consider misuse.

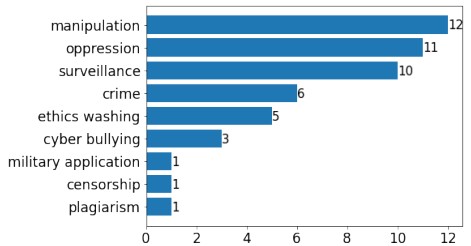

Figure 5: Distribution of codes for harms identified by participants of the tasks in NLP they work on.

| Area | vulnerability | harms |
|---|---|---|
| NLP Applications | 4.3 | crime, oppression, manipulation |
| Ethics and NLP | 4.1 | ethics washing, surveillance, plagiarism, oppression |
| Psycholinguistics | 4.0 | cyber bullying, oppression |
| Generation | 3.8 | crime, cyber bullying, manipulation, oppression, ethics washing |
| Dialogue Systems | 3.8 | surveillance, crime |
| MT and Multilinguality | 3.3 | surveillance, crime |
| ML for NLP | 3.2 | military application, manipulation, oppression |
| Resources and Evaluation | 3.1 | ethics washing, surveillance, manipulation |
| Interpretability | 3.0 | ethics washing, manipulation |
| Information Extraction | 2.8 | surveillance, censorship |
| IR and Text Mining | 2.7 | surveillance |

Table 1: Average score for vulnerability across ACL areas (with at least three answers) the participants work on and their associated harms.

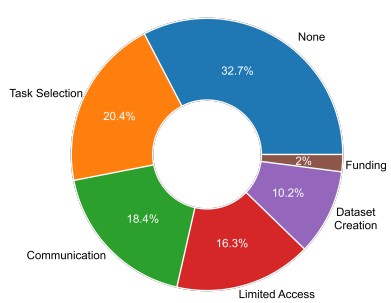

Figure 6: Distribution of the measures participant take to limit misuse of the research artefacts they create.

able each task is to misuse on a Likert scale from 1 to 5, where 5 is *very vulnerable* (see Table 1 for aggregate scores and harms associated with each area). For each area, we find a wide range in harms for associated tasks, thus indicating the importance of considering dual use within specific use-cases.

### 3.4 Prevention of Misuse

The next part of the survey sought to uncover how participants currently mitigate misuse of their research artefacts which we inductively code and aggregate (see Figure 6 for the distribution of coded answers). The largest share of participants (31.8%) stated that they do not take any steps to prevent the misuse of their work. This stands in contrast to participants assessment of the risks of misuse of NLP and the tasks they work on, respectively (see Section 3.3). Participants who do seek to mitigate misuse primarily (22.7%) do so through selecting tasks they believe are not prone to harmful

use. For instance, they highlight factors that they believe are indicative of tasks that could be less prone to dual use, e.g., *"niche tasks which (hopefully) can't be generalised such that they become harmful"*. However, the particular methods for selecting a task which is not prone to dual use, or one that carries risks of it, are not described, indicating a need for a standardised framework to help guide professionals in selecting tasks. Participants also measure other methods for addressing dual use concerns, e.g., including ethics statements and outlining limitations of the research in the paper. While the ethical impacts statements have only recently become mandatory at ACL conferences, they

support researchers in clearly communicating their concerns and intentions when publishing research. Participants also highlight other guidelines, such as data statements (Bender and Friedman, 2018) as helpful in outlining the limitations and risks of their research. Finally, limiting access to scientific artefacts is also used to mitigate misuse. Although this strategy can help mitigate misuse, it can have adverse effects on access for legitimate purposes.

**Institutional Support**   The final questions focus on institutional support for participants when it comes to questions of societal harm. When asked about support from institutions, only 37.5% of participants feel positive about the support they receive (see Figure 3). That is, the majority of participants do not feel adequately supported by their institution in terms of preventing the misuse of their research. Selecting between *Ethics boards, Courses, Training Material*, and *None* most participants selected that they had ethical review boards available (see Figure 3). While 61% of participants have previously used institutional support, 14% of participants stated that they had no access to institutional support for dual-use concerns. Such a lack of support needs to be addressed from all institutional partners, i.e., publication venues, universities, and companies engaged in the production of research artefacts. Until there are structured efforts to prevent the misuse of research, however, the responsibility lies with the professionals themselves. Therefore, we propose a checklist in Section 5 to provide a starting point for researchers to address some of the concerns of dual use.

## 4   Prevention of dual use

In prior work, there are a set of approaches that address the problem of mitigating the misuse of research. In the following, we present four approaches mentioned in the context of machine learning at large; *forbidden knowledge*, ethics review boards, education of researchers, and guidelines/checklists.

### 4.1   Forbidden Knowledge

As knowledge holds power, the term *forbidden knowledge* describes scientific fields that are too dangerous to distribute (Johnson, 1996). In the context of machine learning, Kempner et al. (2011) propose to develop and establish an ethical framework to address this field's forbidden knowledge

and dual use applications. They present a framework to assess forbidden knowledge in machine learning research. They further propose grades of availability for the applications resulting in a research project, i.e., in which way it could be restricted. The authors point out that it is infeasible to halt the development of machine learning in directions that could lead to negative outcomes, as machine learning *"is in most cases a general purpose or dual use technology, meaning that it has general capabilities, which are applicable to countless varying purposes"* (Kempner et al., 2011).

Contrarily, Marchant and Pope (2009) point out the problems with forbidding science, including: the unenforceability of forbidding a direction of research internationally, especially as international agreements are rare; the legislative imprecision in a rapidly evolving field; the unpredictability of which scientific advances may have harmful applications; the outdated nature of laws regulating science; and the potential misuse of such laws for parochial interests. The authors emphasise the challenges of identifying dual use of a research artefact ahead of time, as the main research outcome of an artefact might be beneficial and therefore poses the question of the likelihood of destructive or malicious applications.[9] They argue that instead of forbidding one research direction altogether, regulation should be put in place to forbid the misuse of research. Another option would be to pursue the research direction but limit the publication of the scientific results or the models themselves, however the regulations should come from inside the communities, i.e., self-regulation. Contributing to that discussion, Brundage et al. (2018); Solaiman (2023) discuss strategies for controlled release of AI models as a mitigation method and the associated trade-offs. Going a step further, Henderson et al. (2023) argue for more technical approaches to supplement structural strategies to mitigate risks. They demonstrate an approach for embedding task blocking within a language model.

### 4.2   Ethics Review Boards

In the EU, the *Nuremberg Code* set the precedent for the importance of consent in medical research and later the *Declaration of Helsinki* set the standard for ethical medical research in the EU. Any project funded by the EU undergoes an ethics re-

---

[9]Similar to our work, they differentiate dual use and *morally objectionable* research directions.

view.[10] In the US and other countries, Institutional Review Boards (IRB) are required by law and are appointed to make decisions on ethics with regard to research on human subjects (Grady, 2015).

Bernstein et al. (2021) propose the creation of an *Ethics and Society Review board* (ESR), which intervenes at the stage of the submission of research proposals. They argue that currently established ethics boards are concerned with harm to human subjects rather than impact on society at large. One of the topics that they focus on is dual use, which they define as *"Any risks or concerns that arise due to the technology being co-opted for nefarious purposes or by motivated actors (e.g., an authoritarian government employed mass surveillance methods)"* (Bernstein et al., 2021). They test their proposed ESR board and find that the resulting proposals are influenced by the ESR board's focus on ethics. Within machine learning, IRB or ERB reviews are typically done for projects involving human annotators or subjects but the wider societal impacts of research are rarely reviewed. Recently, pre-registration has been proposed as a method to improve robustness and reliability of machine learning research (van Miltenburg et al., 2021; Søgaard et al., 2023; Bertinetto et al., 2020; Albanie et al., 2021). While improving scientific rigour, pre-registration could also serve as an opportunity to evaluate the ethical and dual use implications of projects at the proposal stage.

### 4.3 Education

Another path to advocating for a more careful interaction with dual use is the possibility of educating researchers about its danger. Minehata et al. (2013) argue for the education of life scientists w.r.t. dual use, starting as part of their university education. This shifts the responsibility towards universities to provide these courses as well as on to the researchers to have been provided with the courses. While integrating the topic of dual use in the curriculum is an important issue, it requires a large, coordinated effort, that is yet to materialise.

With regards to NLP, there have been proposals to integrate ethics into the NLP curriculum, including dual use (Bender et al., 2020; Strube and Pan, 2022). In their proposal, Bender et al. (2020) suggest education on dual use by including *"Learning how to anticipate how a developed technology could be repurposed for harmful or negative results, and designing systems so that they do not inadvertently cause harm."*. We argue that educating researchers about the potential harms of their research should be a core part of their training. However, as a prevention method, since it is limited to people receiving formal research training and does not reach all practitioners creating NLP tools, it should be used in conjunction with other methods for prevention of dual use.

### 4.4 Checklists and Guidelines

A proposal presented in previous studies is the implementation of checklists and guidelines for researchers to navigate the complex issue of ethics in their research area (Madaio et al., 2020b; Rogers et al., 2021). A wide range of checklists have been created in recent years to cover the topic of ethics and fairness in AI, e.g., ARR's Responsible NLP Research checklist.[11], NeurIPS' paper checklist[12], and Microsofts' AI Fairness Checklist[13] However, existing checklists for AI Ethics at large, and for NLP in specific, do not yet cover the topic of dual use. Guidelines such as data statements (Bender and Friedman, 2018) and model cards (Mitchell et al., 2019) encourage researchers to report on their scientific artefacts more thoroughly, document their intended use, and reflect on the impact those can have. Mohammad (2022) present an ethics sheet for AI, which considers the impact and ethics of AI projects. Although dual use is briefly mentioned, it does not cover the different aspects of dual use that are required in order to suffice as a starting point for discussion of researchers in the field. In contrast, the checklist proposed in Section 5 is designed to be concise and specifically focused on dual use, rather than having long-form text answers as in the ethics sheet.

Checklists as a method in AI ethics have been recently criticised due to their representation of modularity of software and the shifting of responsibility away from the creator of the software (Widder and Nafus, 2023). While we attempt to integrate that criticism into the creation of the checklist itself, checklists can only ever be a starting point for a conversation around AI ethics at large and misuse

---

[10]https://ec.europa.eu/research/participants/data/ref/fp7/89888/ethics-for-researchers_en.pdf

[11]https://aclrollingreview.org/responsibleNLPresearch/

[12]https://neurips.cc/Conferences/2021/PaperInformation/PaperChecklist

[13]https://www.microsoft.com/en-us/research/project/ai-fairness-checklist/

of research artefacts in specific. Checklists provide a productive first step to help researchers, who are unfamiliar with an area, to approach and consider it. Therefore, checklists are a productive starting point and should not be the end-goal.

## 5 Checklist for Dual Use

Based on the results of the survey and previously established guidelines and checklists, we propose the following checklist for the issue of dual use in NLP. This checklist should be integrated into existing checklists and guidelines to extend them towards this topic and give researchers a guideline for the issue of malicious reuse of their work. The design of our checklist is inspired by Madaio et al. (2020a), who co-design a checklist for fairness in AI with the input of practitioners. They argue that checklists should formalise existing informal processes. Additionally, our design takes into account the insights of Rosen (2010), who maintain that checklists are essential in complex and specialised fields to ensure that the minimum necessary steps are taken. Based on their experience in medicine, Rosen (2010) found that simple checklists are most effective in reducing human error.

The *dual use of scientific artefacts* part of the checklist can be used at the ideation stage of a research project as well as in the stages before the publication of the paper. We believe that integrating these questions in existing checklists, such as ACL's *Responsible NLP Research Checklist* will draw attention to the topic of dual use and can start a much-needed discussion in the field of NLP on such issues.

### 5.1 Checklist Creation

After collecting the feedback of participants on their perspective of dual use in the survey (Section 3), we grouped and sorted the feedback to be able to extract questions for the checklist that tackles the issues mentioned by participants. We grouped the checklist along the two most important angles for the prevention of misuse as discussed in Section 3.4: the *dual use of scientific artefacts* and *prevention of dual use*. The *dual use of scientific artefacts* part aims to give researchers space to reflect on their work and the potential for misuse. The *prevention of dual use* part aims to make preventive measures explicit and open a discussion about institutional support for the topic of dual use.

**Dual use of scientific artefacts** Despite potential unfamiliarity with the concept of dual use among survey participants, we found that they could identify a range of potential harms deriving from misuse of their work (see Section 3.3). The objective of the first question in the checklist (C1) is to elicit the full range of concerns held by a researcher and ensure they are fully expressed in their paper. The following questions draw from the potential harms survey participants mentioned in Section 3.3 as displayed in Figure 5. We selected from these harms the ones that can be applied to a wide range, leaving out ethics washing (as it is specific to a subset of areas in NLP), crime (as it is too wide of a question and can be covered by the other topics), cyber bullying (as it could be repetitive with oppression), and plagiarism (as it is again covering only a specific field, potentially repretative). This leaves questions about whether scientific artefacts resulting from the research can be used for surveillance (C2), military applications (C3), harm or oppression of (marginalised) groups in society (C4), or to manipulate people by spreading misinformation or polarising users (C5).

**Prevention of dual use** A large share of survey participants stated that they did nothing to prevent misuse of their work (see Section 3.4). Therefore, for the second part of the checklist, we focus on ways to mitigate harm through dual use. The first question in this section (C6) is aimed to cover all possible ways that participants try to prevent harm through dual use. This could be by using institutional support in form of data distribution frameworks or by using licensing options as provided and described by external entities. With the following two questions, we focus on institutional support. We found that while a majority of researchers have and access institutional support (see Section 3.4), a large share of participants are yet to use their institutional support. In Section 4, we identify a number of possibilities to mitigate the malicious reuse of research that can only be implemented on an institutional level, e.g., ethics review boards and education. Towards the goal of better education, we phrase the checklist question C7 about ethics training. Bernstein et al. (2021) propose the extension of ethics review boards to *Ethics and Society Review board*, indicating the importance of institutional support for questions of societal harm. The checklist question C8 covers these concerns, asking whether the scientific artefacts produced were

reviewed for dual use.

## 5.2 Checklist

**Dual use of scientific artefacts**

C1 *Did you explicitly outline the intended use of scientific artefacts you create?*

C2 *Can any scientific artefacts you create be used for surveillance by companies or governmental institutions?*

C3 *Can any scientific artefacts you create be used for military application?*

C4 *Can any scientific artefacts you create be used to harm or oppress any and particularly marginalised groups of society?*

C5 *Can any scientific artefacts you create be used to intentionally manipulate, such as spread disinformation or polarise people?*

**Prevention of dual use**

C6 *Did you access your institution's or other available resources to ensure limiting the misuse of your research?*

C7 *Have you been provided by your institution with ethics training that covered potential misuse of your research?*

C8 *Were the scientific artefacts you created reviewed for dual use and approved by your institution's ethics board?*

## 6 Conclusion

In this paper, we discussed the topic of dual use, the intentional, harmful misuse of research artefacts, in the context of NLP research. Dual use has been overlooked in the field of NLP, a gap this paper aims to close. In order to gauge the current level of consideration for dual use in the field of NLP, we conducted a survey among NLP professionals. Our survey results revealed that a majority of the participants expressed concerns regarding the potential misuse of their work. They identified several potential dangers associated with their work, including the oppression of (marginalised) groups in society, surveillance, and manipulation. However, they reported taking limited steps toward preventing such misuse. Based on the participants' inputs on a provided definition of dual use, we propose a definition of dual use that is more appropriate for researchers in the field of NLP. We discuss existing proposals to address dual use in machine learning and other research fields. Many of these approaches need institutional structures to be realised on a large scale.

Despite the participants' concerns about the potential for misuse of their work in NLP, there appears to be a lack of institutional support for addressing these concerns fully. It is imperative for structured and comprehensive initiatives to be implemented to address this issue. Utilising the participants' insights, we have developed a dual use checklist for NLP professionals. This checklist, which can be integrated with existing ethics checklists, addresses both the potential dual use of scientific artefacts and measures to prevent such dual use. In conclusion, the dual use of NLP artefacts is a critical concern that has received limited attention in the field. Our paper strives to raise awareness and initiate a conversation among NLP professionals to address this important and overlooked issue.

## Limitations

Every study relying on surveys and a mixed-methods approach will have a set of limitations. In the case of our study, we find that the participants of the survey underlie a bias based on the location and network of the leading researchers of this project. While the survey was distributed both through the researchers' Twitter accounts as well as Reddit, which could potentially reach a more diverse community, most answers are from researchers based in North America and Europe (see Section 3.1). This constrains our findings to the issue of dual use as seen through a Western lens, as concerns of dual use may be different across different geographies and social configurations (Revill et al., 2012). It is therefore important to bear in mind that our findings and proposed methods are not necessarily applicable universally and studies that examine the perspective on dual use in other geographies are important for the field of NLP to be able to develop culturally competent methods for addressing dual use. Further, given the topic and name of the survey, it is likely that participants who are interested in the topic of misuse of research are more likely to answer the full survey. However, we do find some researchers who are clearly opposing the underlying premises of the study. Quoting one of the comments at the end of the survey, one participant wrote: *"Please don't ruin science."* pointing out their disagreement with the overall topic of ethics in NLP.

## Acknowledgements

This research was partially funded by a DFF Sapere Aude research leader grant under grant agreement No 0171-00034B, as well as supported by the Pioneer Centre for AI, DNRF grant number P1.

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

# A Codebooks

For each of the free-form answer fields of the survey, we coded the answers of the participants, for details see Section 3. Below we provide the codebooks for the relevant questions reported in this paper.

## A.1 Harms by Task

Participants were asked to list the potential harm of each of the tasks they stated they worked on. The codebook is provided in Table 2. The distribution of the codes among the survey respondents can be found in Figure 5.

[h]

## A.2 Measures to limit misuse

Participants were asked to list measures they take to limit the potential misuse of their work. The codebook is provided in Table 3. The distribution of the codes among the survey respondents can be found in Figure 6.

# B Survey

We provide the full survey as conducted on LimeSurvey below.

# Dual Use - NLP

There are 23 questions in this survey.

## Survey disclaimer

This survey is aimed at researchers and practitioners in the field of natural language processing (NLP). This research is conducted by Lucie-Aimée Kaffee [1], Arnav Arora [1], Zeerak Talat [2], Isabelle Augenstein [1], affiliated with the [1] University of Copenhagen and [2] Digital Democracies Institute with the goal of understanding how researchers approach the danger of misuse of their research.

This survey has been approved by the ethics committee of the University of Copenhagen under number 504-0313/22-5000

We will ask you a mix of free-form text questions and multiple-choice or Likert scale questions. In total, this survey will take a maximum of 20 minutes of your time. We would like to ask you to fill the survey exhaustively and with as much detail as possible.

The survey data is collected anonymised and will be aggregated for publication. We will cite free-form text answers but never in connection with any possibly identifying data. Since the survey is anonymised we cannot offer to withdraw or change answers.

## Background

### Do you work in industry or academia? (If other, please specify in the text box)

\*

❶ Choose one of the following answers
Please choose **only one** of the following:

○ Industry

○ Academia

○ Both

○ Other [                    ]

### In Industry what best describes your position? \*

Only answer this question if the following conditions are met:
((G02Q34.NAOK (/questionAdministration/view/surveyid/987789/gid/1/qid/230) == 'AO02' or G02Q34.NAOK
(/questionAdministration/view/surveyid/987789/gid/1/qid/230) == 'AO05'))

❶ Choose one of the following answers
❶ If you choose 'Other:' please also specify your choice in the accompanying text field.
Please choose **only one** of the following:

○ Research Scientist

○ Research Engineer

○ Industrial postdoc

○ Management

○ Other [                    ]

## In Academia, what best describes your position? *

Only answer this question if following conditions are met:
((G02Q34.NAOK (/questionAdministration/view/surveyid/987789/gid/1/qid/230) == 'AO03' or G02Q34.NAOK
(/questionAdministration/view/surveyid/987789/gid/1/qid/230) == 'AO05'))

❶ Choose one of the following answers
Please choose **only one** of the following:

○ Master student

○ PhD candidate

○ Postdoc

○ Lecturer

○ Assistant/Associate Professor

○ Professor

○ Other [                              ]

## Do you deploy NLP applications into production? *

Please choose **only one** of the following:

○ Yes

○ No

## Where in the world do you work?

*

❶ Choose one of the following answers
Please choose **only one** of the following:

○ Africa

○ Asia

○ Australia/Oceania

○ Antarctica

○ Europe

○ North America

○ South America

Background - fields of research

We want to understand what parts of NLP you have been previously or are currently working on. Below, we provide you with a list of fields in NLP.

Please select below, which fields you have worked in (these are the ACL tracks, imagine which ones you have previously, would have or will submit papers to). Please focus on your primary fields of work.

For each field, please provide a list of tasks that you work on in that field, **separated by commas**.

Tasks can be more specific areas of research, not necessarily problem formulations. For example, tasks in *Generation* could be *multilingual data to text generation, poetry generation*. If the field is the same as the task you work on, just repeat or specify the task in the text field. *

❶ Comment only when you choose an answer.
Please choose all that apply and provide a comment:

| | |
|---|---|
| ☐ Cognitive Modeling and Psycholinguistics | |
| ☐ Computational Social Science and Cultural Analytics | |
| ☐ Dialogue and Interactive Systems | |
| ☐ Discourse and Pragmatics | |
| ☐ Ethics and NLP | |
| ☐ Generation | |
| ☐ Information Extraction | |
| ☐ Information Retrieval and Text Mining | |
| ☐ Interpretability and Analysis of Models for NLP | |
| ☐ Language Grounding to Vision, Robotics and Beyond | |
| ☐ Linguistic Theories, Cognitive Modeling, and Psycholinguistics | |
| ☐ Machine Learning for NLP | |
| ☐ Machine Translation and Multilinguality | |
| ☐ NLP Applications | |
| ☐ Phonology, Morphology, and Word Segmentation | |
| ☐ Question Answering | |
| ☐ Resources and Evaluation | |
| ☐ Semantics: Lexical | |
| ☐ Semantics: Sentence-level Semantics, Textual Inference, and Other Areas | |
| ☐ Sentiment Analysis, Stylistic Analysis, and Argument Mining | |
| ☐ Speech and Multimodality | |
| ☐ Summarization | |
| ☐ Syntax: Tagging, Chunking and Parsing | |

## Dual use definition and tasks

The European Union define dual use as: **"goods, software and technology that can be used for both civilian and military applications"**
As we look specifically into NLP research, we adapt and extend this definition as follows: "**Dual use describes any task that can be intentionally used for a harmful (secondary) purpose beside its main application area.**"

Do you think it is important to have a definition of dual use that can be applied to NLP?
*

Please choose **only one** of the following:

◯ Yes
◯ No

---

Do you think that our definition of dual use (above) is appropriate for NLP? *

Please choose **only one** of the following:

◯ Yes
◯ No

---

How would you adapt our definition of dual use? *

Please write your answer here:

[text field]

---

As you stated previously, you work on the following tasks: {INSERTANS:987789X5X1SQ003comment}
{INSERTANS:987789X5X1SQ004comment}{INSERTANS:987789X5X1SQ005comment}
{INSERTANS:987789X5X1SQ006comment}{INSERTANS:987789X5X1SQ007comment}
{INSERTANS:987789X5X1SQ008comment}{INSERTANS:987789X5X1SQ009comment}
{INSERTANS:987789X5X1SQ010comment}{INSERTANS:987789X5X1SQ011comment}
{INSERTANS:987789X5X1SQ012comment}{INSERTANS:987789X5X1SQ013comment}
{INSERTANS:987789X5X1SQ014comment}{INSERTANS:987789X5X1SQ015comment}
{INSERTANS:987789X5X1SQ016comment}{INSERTANS:987789X5X1SQ017comment}
{INSERTANS:987789X5X1SQ018comment}{INSERTANS:987789X5X1SQ019comment}
{INSERTANS:987789X5X1SQ020comment}{INSERTANS:987789X5X1SQ021comment}
{INSERTANS:987789X5X1SQ022comment} {INSERTANS:987789X5X1SQ023comment}
{INSERTANS:987789X5X1SQ024comment} {INSERTANS:987789X5X1SQ025comment}

For each task you defined previously, there is one text field. If you defined less than 25 tasks, please leave the additional fields empty. Please add for each task the following:

**Dual Use Use Cases**: In the text field, state which use cases of the task could be problematic (e.g., uses when models are *intentionally* deployed out of context or cause harm)

**Task Vulnerability**: On the scale, define how vulnerable is the task to misuse in the form of dual use (e.g., for the use cases you described).

Please choose the appropriate response for each item:

| | Very vulnerable | Somewhat vulnerable | Neither vulnerable nor invulnerable | Not vulnerable | Not vulnerable at all |
|---|---|---|---|---|---|
| **Task 1** | ○ | ○ | ○ | ○ | ○ |
| **Task 2** | ○ | ○ | ○ | ○ | ○ |
| **Task 3** | ○ | ○ | ○ | ○ | ○ |
| **Task 4** | ○ | ○ | ○ | ○ | ○ |
| **Task 5** | ○ | ○ | ○ | ○ | ○ |
| **Task 6** | ○ | ○ | ○ | ○ | ○ |
| **Task 7** | ○ | ○ | ○ | ○ | ○ |
| **Task 8** | ○ | ○ | ○ | ○ | ○ |
| **Task 9** | ○ | ○ | ○ | ○ | ○ |
| **Task 10** | ○ | ○ | ○ | ○ | ○ |
| **Task 11** | ○ | ○ | ○ | ○ | ○ |
| **Task 12** | ○ | ○ | ○ | ○ | ○ |
| **Task 13** | ○ | ○ | ○ | ○ | ○ |
| **Task 14** | ○ | ○ | ○ | ○ | ○ |
| **Task 15** | ○ | ○ | ○ | ○ | ○ |
| **Task 16** | ○ | ○ | ○ | ○ | ○ |
| **Task 17** | ○ | ○ | ○ | ○ | ○ |
| **Task 18** | ○ | ○ | ○ | ○ | ○ |
| **Task 19** | ○ | ○ | ○ | ○ | ○ |
| **Task 20** | ○ | ○ | ○ | ○ | ○ |
| **Task 21** | ○ | ○ | ○ | ○ | ○ |
| **Task 22** | ○ | ○ | ○ | ○ | ○ |
| **Task 23** | ○ | ○ | ○ | ○ | ○ |
| **Task 24** | ○ | ○ | ○ | ○ | ○ |
| **Task 25** | ○ | ○ | ○ | ○ | ○ |

Dual Use definition and considerations

## Overall, as a field, how vulnerable is NLP to dual use?

\*

Please choose the appropriate response for each item:

| | Very vulnerable | Somewhat vulnerable | Neither vulnerable nor invulnerable | Not vulnerable | Not vulnerable at all |
|---|---|---|---|---|---|
| | ○ | ○ | ○ | ○ | ○ |

---

Which, if any, of the scientific artefacts you create or use could be a problem for society or a group of people in a society in terms of dual use?

❶ Check all that apply

Please choose **all** that apply:

- ☐ Methods
- ☐ Data
- ☐ Model
- ☐ Analysis
- ☐ Interface
- ☐ Other: ____________________

---

When, during the course of a research project (if at all), do you consider potential misuses of the technologies or knowledge you create or use? \*

❶ Check all that apply

Please choose **all** that apply:

- ☐ In the idea creation phase
- ☐ When applying for grants
- ☐ When reading related work
- ☐ When exchanging ideas with colleagues
- ☐ When running experiments
- ☐ While writing the paper
- ☐ When submitting the paper
- ☐ When filling in ethical statements/checklists during submission
- ☐ When reading reviews
- ☐ When preparing the camera ready version
- ☐ After publication
- ☐ When the work is cited
- ☐ At none of these stages
- ☐ Other: ____________________

---

## How often do you think of the question of potential misuse of your research?

\*

Please choose the appropriate response for each item:

| | Always | Often | Sometimes | Rarely | Never |
|---|---|---|---|---|---|
| | ○ | ○ | ○ | ○ | ○ |

---

## Which questions do you ask yourself regarding the potential misuse of your research?

How important is the question of potential misuse of your research to you?
*
Please choose the appropriate response for each item:

|  | Very important | Somewhat important | Neither important nor unimportant | Not important | Not important at all |
|---|---|---|---|---|---|
|  | ○ | ○ | ○ | ○ | ○ |

What measures do you take to limit the potential misuse of your research (if any)? *

Please write your answer here:

How supported by your institution do you feel with regards to preventing the misuse of your research?
*
Please choose the appropriate response for each item:

|  | Very supported | Somewhat supported | Neither supported nor unsupported | Not supported | Not supported at all |
|---|---|---|---|---|---|
|  | ○ | ○ | ○ | ○ | ○ |

Which, if any, support for such considerations does your institution provide?
❶ Check all that apply
Please choose **all** that apply:

☐ Ethics boards
☐ Courses
☐ Training material
☐ None
☐ Other:

Have you utilised said support for any of your projects?
Please choose **only one** of the following:

○ Yes
○ No

Do you have any comments for the researchers?

Please write your answer here:

07-04-2022 – 12:01
Submit your survey.
Thank you for completing this survey.

| code | description | includes(sub-codes) | examples |
|---|---|---|---|
| surveillance | automation of surveillance by any entity | surveillance by corporations; surveillance by government | large scale data collection and processing of users |
| manipulation | manipulation of users of technology, trying to change their beliefs or world-view | disinformation, polarisation | disinformation generation |
| oppression of groups in society | technologies to oppress, marginalise or disadvantage any group in a society | language standardisation; propagation of racist believes or classification | generation of hate speech; classification based on dialects in a language |
| crime | reuse of technologies for general criminal purpose | | personalised phishing |
| ethics washing | reuse technologies to minimise the underlying problems of a system | | explainability to fairwash an algorithm exhibiting bias |
| cyber bullying | target individuals with harmful, oppressive or disadvantageous content | | using technologies to bully kids in schools |
| military application | reuse of technologies by the military | | target systems |
| censorship | automation of censorship | censorship by government | finding content to censor on a large scale |
| plagiarism | automatic creation or summarisation of content for plagiarism | | automatically rephrasing existing academic work to republish |

Table 2: Codebook for the harms identified by participants for each of the tasks they work on. Question: *Please add for each task the following: Dual Use Use Cases: In the text field, state which use cases of the task could be problematic (e.g., uses when models are intentionally deployed out of context or cause harm).*

| code | description | includes (sub-codes) | examples |
|---|---|---|---|
| Task Selection | considerations regarding the tasks the participants work on | | selecting tasks or research problems to work on that are less prone to misuse; decline jobs or research projects where participants are not comfortable how their models are used |
| Communication | communicating the risks related through the work; discussing potential risks with the community | ethics statements | writing the paper to clearly communicate risks; using ethics statements |
| Limited Access | limiting access to scientific artefacts | regulation work; licensing | limit access to data and code |
| Dataset Creation | careful considerations when the dataset is created with regards to how it will be reused | dataset statements | anonymisation of datasets; keep processing of data local |
| Funding | considerations regarding the sources of funding for research | | selection and rejecting projects based on the funding agency |
| None | no measures taken to limit malicious reuse | | comment "None" |

Table 3: Codebook for how participants select tasks to work on. Question: *What measures do you take to limit the potential misuse of your research (if any)?.*