# OpenReview forum: "Thorny Roses: Investigating the Dual Use Dilemma in Natural Language Processing"
_EMNLP/2023/Conference — EMNLP 2023 Findings_

### Official Review · Reviewer_chMo · 2023-07-28

**Soundness:** 2

**Excitement:**

3: Ambivalent: It has merits (e.g., it reports state-of-the-art results, the idea is nice), but there are key weaknesses (e.g., it describes incremental work), and it can significantly benefit from another round of revision. However, I won't object to accepting it if my co-reviewers champion it.

**Missing References:**

I do not know af any references that should be included, but that is not the main issue.

**Paper Topic And Main Contributions:**

The paper's topic is about dual use and its risks in NLP. A specific definition of dual use is provided and the paper ends with a dual use checklist. Along the way, empirical results as to the attitudes of members of the NLP community vis-a-vis dual use are provided.

**Questions For The Authors:**

See previous section on this.

**Reasons To Accept:**

The paper makes an original contribution in that the problems associated with dual use of NLP techniques (called 'artefacts' in the paper) have not been properly explored in the relevant literature. This provides a reason to accept the paper.

**Reasons To Reject:**

The paper gives several reasons to reject it. First, it is conceptually rather sloppy. Let me give a few examples to back this assessment. Throughout, authors use the concept of 'harm' in an unproblematised way, defined roughly as 'adverse consequences on someone' (the standard definition here would be 'setback of people's interests'). But this neutral concept of harm notoriously does not give reasons, in itself, to others to stop harming people. Only individuals' LEGITIMATE interests count as 'harm' in this second, reason-giving, way. The difference is huge and reveals a big flaw in the paper. So the mere fact that I shall do something that 'harms' you in the first way does not mean that I do not have a right to do it. I only lack the right if your interest is LEGITIMATE. By not attending to this distinction, the authors blur different things that should be kept separate and misidentify both 'malicious intent' and 'harm'. Second example: the authors do not distinguish between a primary domain of application of an NLP artefact and INTENTIONS that an artefact apply to a primary domain. These, again, are not the same. Third example: the authors do not even strive to identify a moral principle that would make their guidelines and discussion plausible. If, however, someone asks 'why exactly is dual use a problem to begin with?', the authors provide no answer. They take it for granted that it does. But that in itself rests (probalbly) on consequentialist grounds (i.e., that agents should avoid actions that lead to suboptimal - under some criterion of value - consequences). The problem is that this principle is very controversial. Some discussion would help, but the authors do not even seem aware that there is an issue here.

ADDENDUM AFTER REBUTTALS: (a) A legitimate interest is a morally relevant interest, whereas a non-legitimate interest is a morally irrelevant interest. Let me give you some prosaic examples to drive the point home. My interest in gaining profits from my company is a legitimate interest in the sense that I have private property in my company, so that of you interfere with it against my will I have a moral claim against you that you stop, but NOT a legitimate interest in the sense that you can open your own company across the road and 'harm' me through competition. Clearly there are adverse consequences for me in the second scanario, but, in a morally relevant sense, you do not harm me. In the same way, I might be sexually attracted to X and s/he might turn me down. Turning me down can have all sorts of adverse consequences, but is morally irrelevant. In quite the same way, if I do not have a right (say) that information about me in the public domain be used (say) to terminate my employment, precisely because (say) there is no 'privacy' in the publlic domain, then adverse consequences do not amount to harm (in the relevant moral sense). .Since harm in the first sense (adverse consequences) is morally neutral, your argument overshoots. To put it provocatively: there is no moral problem with dual use of NLP-artifacts that merely harms people in the sense of non-morally relevant setback of people's interests.
(b) The difference is important because someone might create without harmful intent an NLP-artifact whose primary domain harms people. Does that the use of that artifact to harm people count as dual use?
(c) Sure, but I have tried to say that it's only 'problematic' in the morally-relevant sense, not the morally-neutral one. Again: just because something bad (i.e. something that sets back my interests) happens to me does not mean that someone 'harmed' me in the morally relevant sense. If that's the case, then taking a stance on why dual-use is wrong (presumably because people's legitimate interests can be set back by using NLP artifacts not designed and/or primarily used to set those interests back0 could help identify a normatively proper checklist.

**Reproducibility:**

N/A: Doesn't apply, since the paper does not include empirical results.

**Reviewer Confidence:**

5: Positive that my evaluation is correct. I read the paper very carefully and I am very familiar with related work.

**Typos Grammar Style And Presentation Improvements:**

There are lots of typos throughout, and the paper would benefit from extended proofing.

---

> ### Author Rebuttal · Authors · 2023-08-29
>
> Regarding the **first example**, we do not discuss whether the researchers should have the “right” to produce harmful artefacts, but that they should consider the consequences of their artefacts on society. They can, consequently, decide that the possible harms are reasonable compared to the positive impact of their applications. We ask the reviewer to clarify what their definition of legitimate interest is, since that would entail whether the harm falls under our definition of dual use. Our work provides an operating definition of a kind of harm that we argue should be given more attention and initiate a discussion of researchers along this topic.
>
> **Second example**: Indeed we do not make that distinction. However, in the context of our discussion, we believe that distinction is irrelevant since the intention of the researcher and the primary domain are documented and released with the artefact. Our work focuses on establishing systematic guardrails and providing the researchers with tools to predict and prevent a malicious actor's use of said artefact.
>
> Regarding the **third example**, the issue of dual use is well-defined and described as problematic in the related literature at large [1,2,3]. Our paper is concerned with adapting the topic of dual use to the domain of NLP, not providing a discussion on whether malicious reuse is problematic from an ethical or moral perspective.
>
> [1] Urbina, Fabio, et al. "A teachable moment for dual-use." Nature machine intelligence 4.7 (2022): 607-607.
>
> [2] Forge, John. "A note on the definition of “dual use”." Science and Engineering Ethics 16 (2010): 111-118.
>
> [3] Miller, Seumas, and Michael J. Selgelid. "Ethical and philosophical consideration of the dual-use dilemma in the biological sciences." Science and engineering ethics 13 (2007): 523-580.

---

### Official Review · Reviewer_AsLm · 2023-07-31

**Typos Grammar Style And Presentation Improvements:** 112
**Soundness:** 3

**Excitement:**

4: Strong: This paper deepens the understanding of some phenomenon or lowers the barriers to an existing research direction.

**Missing References:**

Missing reference on line 354

**Paper Topic And Main Contributions:**

In this paper, the authors discuss to what extent the NLP community is currently combating dual use, the unintended misuse of technology. Based on their latest survey, they argue that the members of the NLP an CL communities are not sufficiently aware of the dangers of dual use and discuss how this issue can be addressed in the future. The primary contribution of this paper is a checklist that can be a useful tool or guide for researchers in the community. Additionally, the topic of this paper is rather novel and therefore, this paper can spark interesting and relevant discussions about the future of NLP at conferences. Finally, the authors also present some insights into the general opinion of the research community (regarding ethics and dual use).

**Questions For The Authors:**

A. I believe the added exclusion of "unintended secondary harms" is rather vague. In most cases, I think it would be these harms that are identified by people with malicious intents and then further refined or channeled into dual use. Would that not mean that the difference between dual use and "unintended secondary harms" lies only in the intentionality?

**Reasons To Accept:**

A. The topic of the paper is highly relevant as it discusses how to prevent unethical (dual) use of NLP technology. This is an issue that is not sufficiently researched.

B. The paper concretely proposes a checklist that can be an important step towards building solid ethical guidelines.

C. The work is well-written and the provided argumentation is clear.

D. Great title

**Reasons To Reject:**

A. The definition of dual use is bound to remain vague in some sense, as it relies on the intention of the creator and the user. Additionally, researchers often fail to see past their own intention and can easily overlook creative ways in which the technology can be misused.

B. The paper is rather long. I believe the core message of the paper could also have been written down in a short paper.

**Reproducibility:**

N/A: Doesn't apply, since the paper does not include empirical results.

**Reviewer Confidence:**

3: Pretty sure, but there's a chance I missed something. Although I have a good feel for this area in general, I did not carefully check the paper's details, e.g., the math, experimental design, or novelty.

---

> ### Author Rebuttal · Authors · 2023-08-29
>
> We thank the reviewer for their effort and insights through the review. We will integrate the feedback in the manuscript.
>
> **A.** To address the reviewer’s point that “*researchers often fail to see past their own intention and can easily overlook creative ways in which the technology can be misused.*” We believe that that is exactly why documentation of the researchers’ intention and standardised protocol in the peer review process is needed. The checklist documents the author’s intended use case and helps them think about ways in which their work could be misused and inculcate mechanisms for protection against them. This process does not ensure that all harms will be considered, but we think it plays an important role in putting systematic deterrents to known malicious use cases and in the identification of similar ones in the future.
>
> **B.** We believe that the space for a long paper was necessary to give the different components of the manuscript (the definition, survey, the checklist, and existing interventions for dual use) their due attention in analysing such a nuanced topic, each of which play an important role in getting a holistic overview of the topic. Further, we wanted to give space to the description of the survey to let the reader follow on how the proposed checklist was created.
>
> ### Questions For The Authors
>
> **A.** Intentionality is precisely the distinction we make between dual use and "unintended secondary harms". We believe intentionality is a core component when it comes to understanding the nature of harms of NLP artefacts. Thus, the devised mitigation methods should explicitly factor in the intentions of a potential malicious actor. Currently, work on ethics in NLP primarily focuses on unintended secondary harms [1,2,3,4]. While these are very important topics to address, it also shows the lack of discussion with regards to dual use.
>
> [1] Weidinger, Laura, et al. "Taxonomy of risks posed by language models." Proceedings of the 2022 ACM Conference on Fairness, Accountability, and Transparency. 2022.
>
> [2] Dev, Sunipa, et al. "On Measures of Biases and Harms in NLP." Findings of the Association for Computational Linguistics: AACL-IJCNLP 2022. 2022.
>
> [3] Hardmeier, Christian, et al. "Proceedings of the 4th Workshop on Gender Bias in Natural Language Processing (GeBNLP)." Proceedings of the 4th Workshop on Gender Bias in Natural Language Processing (GeBNLP). 2022.
>
> [4] Blodgett, Su Lin, et al. "Stereotyping Norwegian salmon: An inventory of pitfalls in fairness benchmark datasets." Proceedings of the 59th Annual Meeting of the Association for Computational Linguistics and the 11th International Joint Conference on Natural Language Processing (Volume 1: Long Papers). 2021.

---

### Official Review · Reviewer_mJ7P · 2023-08-11

**Soundness:** 3

**Excitement:**

3: Ambivalent: It has merits (e.g., it reports state-of-the-art results, the idea is nice), but there are key weaknesses (e.g., it describes incremental work), and it can significantly benefit from another round of revision. However, I won't object to accepting it if my co-reviewers champion it.

**Missing References:**

work that discusses dual use in NLP/related fields:
* Koplin, J.J. Dual-use implications of AI text generation. Ethics Inf Technol 25, 32 (2023). https://doi.org/10.1007/s10676-023-09703-z
* K. Nikolskaia and V. Naumov, "Ethical and Legal Principles of Publishing Open Source Dual-Purpose Machine Learning Algorithms," 2020 International Conference Quality Management, Transport and Information Security, Information Technologies (IT&QM&IS), Yaroslavl, Russia, 2020, pp. 56-58, doi: 10.1109/ITQMIS51053.2020.9322897.
* Peter Henderson, Eric Mitchell, Christopher Manning, Dan Jurafsky and Chelsea Finn. "Self-Destructing Models: Increasing the Costs of Harmful Dual Uses of Foundation Models." AIES 2023.

work that discusses ethics interventions/thinking about dual use in NLP/related fields:
* Widder, D. G., & Nafus, D. (2023). Dislocated accountabilities in the “AI supply chain”: Modularity and developers’ notions of responsibility. Big Data & Society, 10(1). https://doi.org/10.1177/20539517231177620
* MAHFOUD, T., AICARDI, C., DATTA, S., & ROSE, N. (2018). The Limits of Dual Use. Issues in Science and Technology, 34(4), 73–78. https://www.jstor.org/stable/26597992

**Paper Topic And Main Contributions:**

The paper studies dual use in an NLP context. The authors propose a definition of dual use that emphasizes the reuse of technology with intent to harm. They conduct a survey of NLP practitioners that asks about their proposed definition and prompts practitioners to speculate about potential dual use in their research area. They propose an ethics checklist focused on dual use.

**Questions For The Authors:**

A. Can you clarify which type of inductive coding you perform (e.g. is this thematic analysis?) and share any resulting codebook?

B. Can you comment on the choice of a checklist in light of recent critique of checklist-based approaches to ethics? For instance, the critiques of Widder & Nafus [1]:  "asking developers to anticipate every conceivable outcome by diligently following elaborate checklists as if they occupied a view from nowhere (what Gansky and McDonald (2022) call “metadata maximalism”) does not portend a meeting of partial knowledges."

[1] Widder, D. G., & Nafus, D. (2023). Dislocated accountabilities in the  “AI supply chain”: Modularity and developers’ notions of responsibility. Big Data & Society, 10(1). https://doi.org/10.1177/20539517231177620

C. The number of full responses is a small fraction of the total respondents (48 of 256). What areas did partial responses leave unfinished, and would it be possible to use some of the data from these partial responses?

D. What do you see as the key contribution of your definition of dual use, in comparison to existing definitions?

**Reasons To Accept:**

S1. Dual use is an important ethical consideration, and the authors discuss it in depth, including discussing the benefits and disadvantages of several definitions, which could be useful for later work in the area.

S2. The authors discuss the literature around several intervention options for dual use, which may not be well-known to NLP practitioners.

S3. The authors highlight the limited discussion of dual use among NLP practitioners (as reflected in their survey), raising a point for concern.

**Reasons To Reject:**

W1. The proposal of the definition is presented as a key contribution. However, the need for an NLP-specific definition of dual use is not well-motivated. While the authors argue clearly that the specific definitions quoted in section 2 are (respectively) too narrow and too broad, it is not clear that discussion of "dual use" in NLP is limited by either of these definitions. In fact, several recent papers engage with "dual use" in ways that seem to parallel the authors' definition, and the ARR Responsible NLP checklist seems to include several of the identified harms explicitly as examples of "malicious [...] uses":
>Examples of risks include potential malicious or unintended harmful effects and uses (e.g., disinformation, generating fake profiles, surveillance), environmental impact (e.g., training huge models), fairness considerations (e.g., deployment of technologies that could further disadvantage or exclude historically disadvantaged groups), privacy considerations (e.g., a paper on model/data stealing), and security considerations (e.g., adversarial attacks). See discussion in Leins et. al. (2020) as examples.

 While "making the `obvious' explicit" can be a valuable contribution of ethics research, it is not clear to me that this clarifies the discussion around dual use or provides a new perspective.

W2. The work could engage more deeply with the STS and ethics literature outside of NLP. Particularly, more discussion of work on how people in technical fields apply or interpret ethics research would be helpful contextualization-- e.g., how do people generally take concepts like "dual use" into NLP ethics work? Does this demonstrate the need for an emic definition of the term, as you suggest?

W3. Overall, the work is interesting descriptively, but could provide more detailed analysis. Particularly, participants indicate that they do not use institutional support and do not explicitly aim to prevent dual use, but without more questions/analysis on *why* participants feel this way, it is not clear that a checklist solution would have any benefits.

W4. Relatedly, the complicated ethical questions around dual use are challenging to address in a survey; long-form interviews may be more appropriate for studying this issue. However, I do appreciate that the authors used some free-response questions in the survey.

**Edit after rebuttal**: the authors have addressed my concern about the contribution (W1), and thus I have raised my excitement score (2->3).

**Reproducibility:**

4: Could mostly reproduce the results, but there may be some variation because of sample variance or minor variations in their interpretation of the protocol or method.

**Reviewer Confidence:**

3: Pretty sure, but there's a chance I missed something. Although I have a good feel for this area in general, I did not carefully check the paper's details, e.g., the math, experimental design, or novelty.

**Typos Grammar Style And Presentation Improvements:**

Line 354: broken reference

Because of the large images (screenshots) in the appendix, the PDF file size is quite large.

---

> ### Author Rebuttal · Authors · 2023-08-29
>
> We thank the reviewer for their comments and their suggestions for related work, we want to take this moment to clarify some of our positions.
>
> **W1.** We believe that a NLP specific definition of dual use is necessary due to how language and identity are co-constructive. This requires specific considerations around language. Pre-existing definitions for dual use are often too narrow or too broad for the purposes of NLP (e.g., regulation). We argue that the general definitions are not adequate to cover the potential misuses, we, therefore, propose a definition tailored with NLP artefacts in mind.  We appreciate the reference to Leins et al., however, their discussion does not address in detail the issue of dual use, but consider it from a higher and more abstract position. We seek to address this with our paper. We further contribute to the understanding of dual use by practitioners through the survey as well as provide an actionable checklist, which enables researchers to reflect on the issue.
> We will be more clear about this in the camera ready.
>
> **W2.** To the best of our knowledge, there isn’t work on mitigating dual use in NLP within those fields. We believe that this further provides evidence for the need to address the concern of dual use. However, the goal of our work was to 1) identify how practitioners view issues of dual use, 2) how they are supported within their institutions, and 3) what might be productive paths forward for considering dual use. We do agree that STS and other social scientific fields have many relevant considerations to NLP, however, it is out of scope for our work, as we are seeking first to establish how the issue is viewed internally.
>
> **W3.** To understand *why* a particular phenomenon exists, we have to first establish that it does, which is what we aim to do with the survey. The survey highlights the stance of the field and builds a foundation for further investigation of the questions posed by dual use in NLP. We believe that future work should address these questions with qualitative and mixed-methods studies to get an in-depth understanding, e.g., of the lack of institutional support, which depends on the context of the individual.
>
> **W4.** We thank the reviewer for the appreciation of our mixed-methods approach. As stated earlier, we wanted to build a foundation for the research in the field of dual-use and NLP and believe that a general overview, including surveys, will build this foundation upon which future qualitative research can build. We would like to suggest that these qualitative studies should be done with the context of the participants in mind, and can then deliver more fine-grained understandings of their needs in their specific context, whereas we wanted to give an introduction to the topic at large.
>
> ###  Questions For The Authors
> **A.** We did indeed use thematic analysis for the coding of the responses. We have added the codebooks at the bottom of the response as a table and will add them to the appendix of the manuscript.
>
> **B.** One of the primary reasons we chose to use a checklist was to integrate our work in existing frameworks around ethics in NLP and not introduce a new format to researchers. We believe that based on discussion within the community, as a result of this paper, this could be extended to other formats in the future, similar to an ethics statement. And while we agree with the critique, checklists provide a productive first step to help researchers, who are unfamiliar with an area, to think about it. We therefore believe that checklists are a productive starting point and should not be the end-goal. We will add these points to the camera ready to encourage future work to develop new formats.
>
> **C.** The partial responses were mostly limited to the participant demographics questions (the first part of the survey), we therefore decided not to use the answers in our analysis of the survey. We will clarify this in the manuscript.
>
> **D.** The main contribution of our definition of dual use is that it is developed specifically with NLP artefacts in mind. Existing definitions of dual use were developed for civilian applications used by the military [1]. In contrast, ours is shaped for the exact scope of the research output of NLP researchers.
>
> [1] Selgelid, Michael J. "Governance of dual-use research: an ethical dilemma." Bulletin of the World Health Organization 87 (2009): 720-723.
>
> ### Codebook for Harms Identified (Figure 5)
>
> |code            |description                                                                                    |includes (sub-codes)      |examples                                                                                                                                                                         |
> |----------------|-----------------------------------------------------------------------------------------------|--------------------------|---------------------------------------------------------------------------------------------------------------------------------------------------------------------------------|
> |surveillance    |automation of surveillance by any entity                                                       |surveillance by corporations; surveillance by government|large scale data collection and processing of users                                                                                                                              |
> |manipulation    |manipulation of users of technology, trying to change their believes or world-view             |disinformation, polarisation|disinformation generation                                                                                                                                                        |
> |opression of groups in society|technologies to oppress, marginalise or disadvantage any group in a society                    |language standardisation; propagation of racist believes or classification|generation of hate speech; classification based on dialects in a language                                                                                                        |
> |crime           |reuse of technologies for general criminal purpose                                             |                          |personalised phishing                                                                                                                                                            |
> |ethics washing  |reuse technologies to minimise the underlying problems of a system                             |                          |explainability to fairwash an algorithm exhibiting bias                                                                                                                          |
> |cyber bullying  |target individuals with harmful, oppressive or disadvantageous content                         |                          |using technologies to bully kids in schools                                                                                                                                      |
> |military application|reuse of technologies by the military                                                          |                          |target systems                                                                                                                                                                   |
> |censorship      |automation of censorship                                                                       |censorship by government  |finding content to censor on a large scale                                                                                                                                       |
> |plagiarism      |automatic creation or summarisation of content for plagirarism                                 |                          |automatically rephrasing existing academic work to republish                                                                                                                     |
>
>
>
> ### Codebook for Task Selection (Figure 6)
>
> |code            |description                                                                                    |includes (sub-codes)      |examples                                                                                                                                                                         |
> |----------------|-----------------------------------------------------------------------------------------------|--------------------------|---------------------------------------------------------------------------------------------------------------------------------------------------------------------------------|
> |Task Selection  |considerations regarding the tasks the participants work on                                    |                          |selecting tasks or research problems to work on that are less prone to misuse; decline jobs or research projects where participants are not comfortable how their models are used|
> |Communication   |communicating the risks related through the work; discussing potential risks with the community|ethics statements         |writing the paper to clearly communicate risks; using ethics statements                                                                                                          |
> |Limited Access  |limiting access to scientific artefacts                                                        |regulation work; licensing|limit access to data and code                                                                                                                                                    |
> |Dataset Creation|careful considerations when the dataset is created with regards to how it will be reused       |dataset statements        |anonymisation of datasets; keep processing of data local                                                                                                                         |
> |Funding         |considerations regarding the sources of funding for research                                   |                          |selection and rejecting projects based on the funding agency                                                                                                                     |
> |None            |no measures taken to limit malicious reuse                                                     |                          |comment "None"                                                                                                                                                                   |

---

### Meta-Review · Area_Chair_WgME · 2023-09-19

**Recommendation:** 4

**Metareview:**

In this paper, the authors formulate an NLP specific definition for the dual use and propose a checklist to be reflected on in ethical considerations of NLP tools. This checklist can be used in addition to a datasheet or model card or other related artefacts to make a complete assessment. The work identifies a major gap in our current discourse of harmful impact of language technologies, and lays foundational work towards defining and checking for the same.
The survey of the NLP practitioners is particularly interesting as a contribution and helps reflect on the state of models being released. The checklist however seems a little under detailed. Some more thought on what one can do with the answers recorded, as well as the agency and knowledge of an NLP researcher to reliably answer them are also points that could be reflected upon in the paper some more, since the checklist is also counted by the authors as a core contribution.
All reviewers in their reviews and comments post rebuttal concur that the paper makes foundational progress towards consideration of dual use in NLP, and with the minor adjustments recommended (such as expansions of terms such as 'secondary unintended harms', image and text typos, mistyped citations in line 354) will consist of significant insights for the community.

---

### Decision · Program_Chairs · 2023-10-07

**Decision:**

Accept-Findings

**Comment:**

In this paper, the authors formulate an NLP specific definition for the dual use and propose a checklist to be reflected on in ethical considerations of NLP tools. This checklist can be used in addition to a datasheet or model card or other related artefacts to make a complete assessment. The work identifies a major gap in our current discourse of harmful impact of language technologies, and lays foundational work towards defining and checking for the same.
The survey of the NLP practitioners is particularly interesting as a contribution and helps reflect on the state of models being released. The checklist however seems a little under detailed. Some more thought on what one can do with the answers recorded, as well as the agency and knowledge of an NLP researcher to reliably answer them are also points that could be reflected upon in the paper some more, since the checklist is also counted by the authors as a core contribution.
All reviewers in their reviews and comments post rebuttal concur that the paper makes foundational progress towards consideration of dual use in NLP, and with the minor adjustments recommended (such as expansions of terms such as 'secondary unintended harms', image and text typos, mistyped citations in line 354) will consist of significant insights for the community.